# Phytochemical Cue for the Fitness Costs of Herbicide-Resistant Weeds

**DOI:** 10.3390/plants12173158

**Published:** 2023-09-02

**Authors:** Hong-Yu Li, Yan Guo, Bo-Yan Jin, Xue-Fang Yang, Chui-Hua Kong

**Affiliations:** 1College of Resources and Environmental Sciences, China Agricultural University, Beijing 100193, China; lihongyulq2011@163.com (H.-Y.L.); guoyan990125@163.com (Y.G.); jinby0505@163.com (B.-Y.J.); 2College of Life Science, Hebei University, Baoding 071000, China

**Keywords:** fitness costs, herbicide resistance, (–)-loliolide, relative fitness, weeds species and biotypes, vitality and fecundity

## Abstract

Despite increasing knowledge of the fitness costs of viability and fecundity involved in the herbicide-resistant weeds, relatively little is known about the linkage between herbicide resistance costs and phytochemical cues in weed species and biotypes. This study demonstrated relative fitness and phytochemical responses in six herbicide-resistant weeds and their susceptible counterparts. There were significant differences in the parameters of viability (growth and photosynthesis), fecundity fitness (flowering and seed biomass) and a ubiquitous phytochemical (–)-loliolide levels between herbicide-resistant weeds and their susceptible counterparts. Fitness costs occurred in herbicide-resistant *Digitaria sanguinalis* and *Leptochloa chinensis* but they were not observed in herbicide-resistant *Alopecurus japonicas*, *Eleusine indica*, *Ammannia arenaria*, and *Echinochloa crus-galli*. Correlation analysis indicated that the morphological characteristics of resistant and susceptible weeds were negatively correlated with (–)-loliolide concentration, but positively correlated with lipid peroxidation malondialdehyde and total phenol contents. Principal component analysis showed that the lower the (–)-loliolide concentration, the stronger the adaptability in *E. crus-galli* and *E. indica*. Therefore, not all herbicide-resistant weeds have fitness costs, but the findings showed several examples of resistance leading to improved fitness even in the absence of herbicides. In particular, (–)-loliolide may act as a phytochemical cue to explain the fitness cost of herbicide-resistant weeds by regulating vitality and fecundity.

## 1. Introduction

Fitness is defined as the ability of a viable and fertile individual to survive certain environments and pass on its genes to offspring, which is an important component affecting the evolution of species in stressed environments [1,2,3]. Evolutionary theory predicts that the adaptation to a new environment will often have negative pleiotropic effects on fitness in the original environment, that is, a so-called “fitness cost” [4,5,6]. The fitness cost is a key component that can influence the evolution of resistance [7,8]. In particular, relative fitness plays a crucial role in rapid resistance evolution. Under selection, the fitness advantage of a resistance trait likely exceeds any fitness cost, leading to rapid resistance evolution. But, when selection is relaxed, the relative fitness of resistant versus wild–type individuals will determine the evolutionary trajectory of resistance genotypes in the absence of selection [9,10].

The fitness costs associated with antibiotic and insecticide resistance have frequently been reported [11,12,13,14]; however, the fitness costs associated with herbicide-resistance genes are not universal [15,16,17]. Herbicide-resistant weeds, a constraint on the economy and productivity of cropping systems, are escalating in farmlands throughout the world [7,18]. There is a lack of information on the differences in relative fitness between herbicide-resistant and -susceptible biotypes and the plant regulatory mechanisms (physiological or genetic) which cause the expression of fitness cost traits. Such information will have theoretical and practical implications for predicting the evolutionary dynamics of weeds resistance.

Fitness is primarily measured by individual reproduction, i.e., fecundity fitness [3,14]. In herbicide-resistant weeds, seed production is used as a crucial determinant to evaluate the relative fecundity fitness [3,6,16,19]. However, seed production may be influenced by other variations in the whole life-history stages, such as phenotypic profiles and physiological and biochemical traits [1,20]. For example, a lower relative growth rate resulted in less seed production in herbicide-resistant weeds, as enzyme activities affect plant growth and fitness [3,15]. Therefore, viability fitness is necessary for assessing the fitness cost of herbicide-resistant weeds. Although the fitness for herbicide-resistant weeds has been reported [21,22,23], herbicide-resistant and -susceptible individuals from different plant populations probably exhibit genetic variability in fitness-related traits, which, consequently, limit our understanding of the fitness cost for resistant biotypes [21,24]. Therefore, in order to unequivocally attribute fitness costs to herbicide resistance, relative fitness should be compared between resistant and susceptible individuals that share the same genetic background.

Although the fitness cost of resistance and its mechanisms in herbicide-resistant weeds have been investigated and interpreted from molecular, physiological, and biochemical perspectives [17,22,25,26,27,28], few studies, to our knowledge, involve phytochemical cues. In fact, phytochemicals, either endogenous hormones or exogenous signals and allelochemicals, play vital roles in plant growth, defense, and reproduction [29,30,31,32]. In particular, phytochemical signals can regulate the tradeoff between growth and defense and aboveground and belowground performance, and thus alter plant fitness [33]. Recent studies have shown that (–)-loliolide, a carotenoid metabolite, is a ubiquitous phytochemical cue impacting plant coexistence and population establishment [31,32,33,34,35,36,37]. (–)-Loliolide can modulate both belowground defense and aboveground flowering in plants [31]. Importantly, (–)-loliolide is a general signal of plant stress that can serve both exogenous and endogenous roles [32,37]. Accordingly, it is reasonable to speculate that (–)-loliolide would be a phytochemical cue in the plant fitness of herbicide-resistant weeds by regulating vitality and fecundity.

The objective of this study was to identify the phenotypic, physiological and biochemical differences between herbicide-resistant and -susceptible weeds, as well as their correlation with fitness, and to test whether there was a fitness cost in the resistant weeds, and whether the fitness cost in herbicide-resistant weeds was associated with variations of (–)-loliolide. To do this, we used six pairs of herbicide-resistant versus -susceptible weeds that commonly occur in rice and wheat fields as model systems. The morphological, biochemical, physiological, and reproductive index between resistant and susceptible biotypes were characterized in the absence of herbicides. Furthermore, (−)-loliolide was quantified in each of the weed species and biotypes, and the relationships between the relative fitness and the variations of (–)-loliolide were analyzed.

## 2. Results

### 2.1. Morphological Traits of Herbicide-Resistant and -Susceptible Weeds

There were different morphological traits between herbicide-resistant and -susceptible biotypes regardless of weed species. Moreover, the differences were varied with the growth stages, especially at the flowering stage (Figure 1 and Appendix A). The plant height, shoot, and root biomass of *L. chinensis* and *D. sanguinalis* were significantly lower in resistant biotypes than in susceptible biotypes, while the opposite was observed in the other weeds except for height for *A. arenaria* and *E. indica* (Figure 1a–c). Significant differences between resistant and susceptible biotypes also occurred in root measurements with an exception of *L. chinensis* (Figure 2 and Appendix A). Compared with susceptible biotypes, resistant biotypes of *A. arenaria* and *D. sanguinalis* had lower total root length (Figure 2a), total root area (Figure 2b), and root volume (Figure 2c). However, these root measurements were significantly greater in resistant *A. japonicus*, *E. crus-galli*, and *E. indica* than in their susceptible biotypes (Figure 2).

### 2.2. Physiological and Biochemical Traits of Herbicide-Resistant and -Susceptible Weeds

The photosynthetic gas-exchange parameters and chlorophyll contents were significantly different between herbicide-resistant and susceptible biotypes (Figure 3). Resistant *A. arenaria* and *A. japonicus* had a higher photosynthetic rate (*Pn*) (Figure 3a), stomatal conductance (*Gs*) (Figure 3b) and transpiration rate *(Tr*) (Figure 3c), and chlorophyll content (Figure 3d) than susceptible biotypes. An opposite trend was observed for *D. sanguinalis* and *L. chinensis* biotypes with the exception of *Tr* in *D. sanguinalis*. The *Pn*, *Gs*, and *Tr* of resistant *E. indica* was significantly higher than those of its susceptible counterpart but the opposite was observed for chlorophyll contents. No statistically significant difference in *Gs* and chlorophyll contents was found between resistant and susceptible *E. crus-galli* (Figure 3b,d), while lower *Pn* and *Tr* occurred in susceptible *E. crus-galli* (Figure 3a,c).

The herbicide resistance of weeds significantly altered antioxidant enzyme activities in shoots and roots (Figure 4). Generally, the plant antioxidative defense enzymes superoxide dismutase (SOD) and catalate (CAT) activities, and phenol concentrations in resistant *A. arenaria*, *A. japonicus*, *E. crus-galli*, and *E. indica* were similar or greater than those in susceptible counterparts (Figure 4a–c). However, herbicide-resistant *D. sanguinalis* and *L. chinensis* had lower SOD and CAT activities and phenol concentrations than the susceptible biotypes. An opposite trend was observed for the lipid peroxidation malondialdehyde (MDA) content in resistant and susceptible weeds (Figure 4d).

### 2.3. Fecundity Performance of Herbicide-Resistant and Susceptible Weeds

Flowering time and seed biomass showed that herbicide resistance regulated the fecundity performance of the weeds. Resistant *A. arenaria*, *A. japonicus* and *E. crus-galli* flowered 5–17 days earlier than their susceptible biotypes. By contrast, resistant *D. sanguinalis* and *L. chinensis* bloomed 2 and 13 days later than susceptible biotypes, respectively (Figure 5a). The hundred kernel weight (HKW) of herbicide-resistant *A. arenaria*, *A. japonicus* and *E. crus-galli* was, respectively, 2.1-, 1.6- and 1.5-fold that of their susceptible counterparts, while herbicide-resistant *D. sanguinalis* and *L. chinensis* had 65.37% and 56.15% lower HKW than susceptible biotypes (Figure 5b). However, herbicide-resistant and susceptible *E. indica* had similar flowering time and seed biomass (Figure 5).

### 2.4. (–)-Loliolide Concentration and Its Relationship with Relative Fitness of Herbicide-Resistant and Susceptible Weeds

The relationships among the morphological, physiological, and biochemical indices of resistant and susceptible weeds were analyzed. In resistant weeds, (–)-loliolide concentration was significant negatively correlated with weed morphological characteristics and photosynthetic gas exchange parameters. However, MDA content and total phenolic content had a significant positive correlation (Figure 6a). Morphology and physiology were negatively correlated in susceptible weeds, but this was not significant (Figure 6b). However, the negative correlation between the photosynthetic rate of susceptible weeds and (–)-loliolide concentrations was significant (Figure 6).

Compared with herbicide-susceptible biotypes, resistant *A. arenaria*, *A. japonicus*, and *E. crus-galli* had higher (–)-loliolide concentration, while resistant *D. sanguinalis* and *L. chinensis* had lower (–)-loliolide concentration. Resistant and susceptible *E. indica* biotypes had similar (–)-loliolide concentration (Figure 7a). Furthermore, the (–)-loliolide concentration was negatively associated with flowering time in both resistant and susceptible weeds (resistant: r^2^ = 0.47, *p* = 0.03; susceptible: r^2^ = 0.66, *p* < 0.001) (Figure 7b,c).

The relative fitness of the resistant and susceptible weeds was estimated. Fitness costs (relative fitness < 1) in term of the relative viability fitness (Figure 7b) and relative fecundity fitness (Figure 7c) were observed in *D. sanguinalis* and *L. chinensis*. On the contrary, a fitness advantage (relative fitness > 1) was observed in resistant *A. arenaria*, *A. japonicus*, *E. crus-galli*, and *E. indica*. The relative fitness was negative correlated with the ratio of (–)-loliolide concentration in resistant weeds and in their susceptible counterparts (Figure 7b,c).

### 2.5. Comprehensive Evaluation of Ecological Adaptability of Herbicide-Resistant and -Susceptible Weeds

The principal component analysis (PCA) clearly distinguished the weeds’ adaptability. The first principal component (PC1 = 55.63.0%) and second principal component (PC2 = 19.71%) together accounted for 75.34% of the total variation (Figure 8a). The results of the PCA indicated a correlation between variables, and a composite score model of ecological adaptability was obtained (Appendix A). The scores of the PCA were closely related to (–)-loliolide concentration; the lower the (–)-loliolide concentration, the higher the integrated score (Figure 8b,c, Appendix A). According to our results, *E. crus-galli* and *E. indica* had greater ecological adaptation.

## 3. Discussion

Evolutionary biologists have often stated that herbicide resistance may lead to reduced fitness costs [15,21,38,39]. In the present study, we evaluated the relative fitness of six herbicide-resistant weeds that commonly occur in paddies and wheat fields against their susceptible counterparts with a similar genetic background, and found that herbicide resistance is not consistently associated with fitness costs. We also found that (–)-loliolide, a ubiquitous phytochemical cue, may play a role in herbicide-resistant relative fitness.

Phenotypic plasticity and viability are important determinants of the fitness for herbicide-resistant weeds. Their effects on fitness ranged from advantage to costs [38,39,40,41,42,43]. In this study, we found that herbicide-resistant *A. arenaria*, *A. japonicus*, *E. crus-galli*, and *E. indica* exhibited great morphological advantages over their susceptible biotypes (Figure 1, Figure 2, Appendix A). Gaines et al. [44,45] also reported that fitness advantage was obvious in glyphosate-resistant *Amaranthus palmeri*, which exhibited high EPSPS gene expression. It may indicate that fitness advantage was observed regardless of herbicide presence or absence for resistant *Amaranthus palmeri*. On the contrary, lower phenotypic traits were observed in herbicide-resistant *D. sanguinalis* and *L. chinensis* than that in the susceptible biotype. Similarly, a nearly lethal fitness cost, manifested as lower plant biomass and growth rate, was observed in a target-site α-tubulin mutation of *Lolium rigidum* [3]. Inhibition of the transport of photosynthetic products and accumulation of ROS could help to explain the lower biomass of weeds [46,47,48,49]. In the current study, the CO_2_ assimilation in mesophyll cells was reduced in herbicide-resistant *D. sanguinalis* and *L. chinensis*, due to lower *Pn*, *Gs*, and *Tr* (Figure 3). Furthermore, lower chlorophyll contents occurred in these two weeds, indicating that chlorophyll content and stomatal factors jointly affected the *Pn* of resistant *D. sanguinalis* and *L. chinensis* (Figure 3d). Protective enzyme activities and MDA contents reflect the degree of damage to the plant cell membrane [50,51,52]. This study demonstrated that lower protective enzyme activities directly increased the accumulation of ROS in herbicide-resistant *D. sanguinalis* and *L. chinensis*, with membrane lipid peroxidation and MDA formation, and may result in fitness costs by decreasing the growth of resistant biotypes (Figure 4).

Flowering marks the transition from vegetative to reproductive growth, which is of great significance in plant fitness [31,53,54]. Early-flowering plants are favored in terms of reproductive success [55]. Resistance plays a role in the evolution of the flowering and reproductive traits of herbicide-resistant weeds [30]. In the current study, herbicide-resistant *A. arenaria*, *A. japonicus*, and *E. crus-galli* accelerated flowering relative to those susceptible biotypes, and improved seed biomass (Figure 5). However, herbicide-resistant *D. sanguinalis* and *L. chinensis* with delayed flowering exhibited poor fecundity when compared with their susceptible biotypes, which led to significant fitness costs at that reproductive stage.

Plants can biosynthesize specialized metabolites to regulate their growth, defense and reproduction. For example, defensive metabolites reduce the performance of pathogens, insects, and competing plants [29,56]. Signaling chemicals trigger a series of responsive strategies with both external and internal hormonal functions in plants [33]. Thus, these specialized metabolites are strongly linked with plant survival and fitness. An increasing number of studies have shown that a ubiquitous phytochemical, (–)-loliolide, is a cross-kingdom metabolite that has a wide range of biological and ecological effects with both exogenous and endogenous activities [32,37]. This study clearly demonstrated that (–)-loliolide influenced the morphological and physiological characteristics of weeds (Figure 6 and Figure 7). (–)-Loliolide stimulates the accumulation of metabolites involved in plant defenses [32,34,35,57] and regulates plant flowering and reproduction [31]. Furthermore, (–)-loliolide plays an important role in kin recognition among biotypes of herbicide-resistant and -susceptible *E. crus-galli* [30]. However, whether (–)-loliolide acts as a phytochemical cue in terms of relative fitness has not been considered yet. This study clearly demonstrated that all weed species tested biosynthesize (–)-loliolide but there was differential (–)-loliolide levels between herbicide-resistant and -susceptible biotypes. According to the results of principal component analysis, *E. crus-galli* and *E. indica* showed lower (–)-loliolide but greater ecological adaptation (Figure 8, Appendix A). (–)-Loliolide as a phytochemical cue is essential to predict a weed’s ecological adaptation and fitness [1,17,32]. In addition, the difference of (–)-loliolide between resistant and susceptible biotypes negatively correlated with plant relative fitness in term of vegetative and reproductive growth, indicating a fitness cost when the concentration of (–)-loliolide in a resistant biotype was higher than its counterpart.

In conclusion, this study focused on determining the universality and mechanism of fitness costs associated with herbicide resistance. We found several examples of resistance leading to improved fitness even in the absence of herbicides. In particular, our study highlights a novel linkage between herbicide resistance costs and phytochemical cues in which the variation in aubiquitous phytochemical (–)-loliolide may help to explain the relative fitness of herbicide-resistant weeds. This study would provide new insight into the evolution of herbicide-resistant weeds by regulating the synthesis of (–)-loliolide. Of course, such (–)-loliolide-based plant fitness detection still requires further verification in other plant systems.

## 4. Materials and Methods

### 4.1. Plant Materials, Soil and Chemicals

Six herbicide-resistant weeds and their susceptible biotypes with the same genetic background were used in the study. Fenoxaprop-p-ethyl-resistant *Alopecurus japonicus*, glyphosate-resistant *Digitaria sanguinalis*, and glyphosate-resistant *Eleusine indica* seeds were collected from wheat fields in Anhui, Hunan, and Sichuan Provinces, China. Bensulfuron-resistant *Ammannia arenaria*, metamifop-resistant *Echinochloa crus-galli*, and cyhalofop-resistant *Leptochloa chinensis* seeds were collected in paddies from Jiangsu Province of China (Appendix A). Herbicide-resistant and -susceptible seeds were identified from a single population with corresponding herbicide treatments for several consecutive years in pot-culture experiments. Briefly, weed seedlings were treated with twice the field recommended dose of their respective herbicide. The surviving individuals were separately surrounded by pollen-proof enclosures to ensure no crosspollination and transplanted to plastic pots (30 cm diameter × 10 cm depth) and grown to maturity. The harvested seeds of each individual were stored in separate paper bags for next-generation selection. In the next year, the harvested seeds of each individual were germinated and transplanted into the plastic pots. At the four- to five-tiller stage, three tillers were carefully cut from the main stem for each plant. The tillers and main stem were separately transplanted into the plastic pots. After 6 days, the tiller individuals were treated with herbicides at varying doses (1/2X, X, 2X and 4X, where X represents the field recommended dose of the appropriate herbicide for each weed). The main stems were not treated with the herbicides. Three weeks after treatment, the plants were classified as a herbicide-resistant biotype when their tillers survived the treatment of the highest dose, while the plants were classified as herbicide-susceptible biotype when the tillers did not survive the treatment of the lowest dose. The corresponding untreated main stem plants of herbicide-resistant and -susceptible biotypes were bagged individually pre-flowering, and seeds were separately harvested from these main stem plants after maturity [58]. Based on a 4-year experiment of continuous selection with the procedure of whole-plant bioassays, the F_4_ homozygous seeds of herbicide-resistant and -susceptible biotypes for each weed species were obtained. This method of screening and stabilizing resistant weeds is widely accepted [16,30,58,59].

Soils were collected randomly from the surface (0–10 cm) of a farm at the Shangzhuang Experimental Station of China Agricultural University (Beijing, China). The soil is a Hapli-Udic Cambisol (FAO classification) with a pH of 6.04, organic matter 26.81 g·kg^−1^, total nitrogen 1.72 g·kg^−1^, available phosphorus was 31.54 g·kg^−1^ and available potassium was 58.96 mg·kg^−1^. Soil samples were air-dried, sieved (2 mm mesh) to remove stones and plant rhizomes, and used in the series of experiments.

(–)-Loliolide was isolated and identified from root exudates using a previously developed method [34], and its authentic standard was obtained from Yuanye Biology Corporation (Shanghai, China). Other organic solvents and chemicals were purchased from China National Chemical Corporation (Beijing, China).

### 4.2. Experimental Design

A pot-culture experiment was conducted in a greenhouse at 20–30 °C night and daytime temperatures and 65–90% relative humidity maintained from May to September 2021. The experiment was conducted in a completely randomized design with sixteen replicates for each herbicide-resistant and -sensitive biotype of six weed species. The seeds for all replicates were from the same parental individual. Surface-sterilized weed seeds were separately sown in Petri dishes (9 cm diameter) with moistened filter paper for pre-germination in a chamber set at a temperature of 28 °C. One pregerminated seed of herbicide-resistant or -susceptible biotype was sown uniformly in the center of each pot (12 cm diameter × 10 cm depth) containing 1000 g soil. All pots were placed in the greenhouse, watered daily and their positions randomized weekly.

Two biotypes of each weed species in 1/4 of pots were sampled at the seedling and tillering stage, respectively. Root morphology was scanned at the seedling stage, while plant height and aboveground and belowground biomass were determined at both stages. Flowering time, photosynthetic parameters, total phenols, antioxidant enzymes, and the malondialdehyde content (MDA) in leaves and roots were measured at the flowering stage in the remaining 1/4 of pots, and seed biomass (hundred kernel weight, HKW) was measured in mature stage in the last 1/4 of pots. (–)-Loliolide was quantified at the flowering stage for each plant as described below.

### 4.3. Phenotypic Profiles of Herbicide-Resistant and -Susceptible Weeds

Plant height was measured with tape. The roots were harvested at the seedling stage, then rinsed carefully and scanned; the images were processed with WinRHIZO software (Regent Instruments Inc., Quebec City, QC, Canada) to obtain three root measurements, two size-related metrics (total root length and total root volume) and a measurement of habitat occupancy (total root surface area) [57]. Finally, the shoots and roots were freeze-dried for biomass determination. The number of days from sowing to the first flower appearance was recorded as the flowering time. Seeds at the mature stage were harvested and their dry weight was recorded. In this study, the sampling time for resistant and susceptible biotypes was consistent within the same weed species, while it was different between weed species owing to varying growth rates.

### 4.4. Photosynthetic Parameters

Photosynthetic gas exchange: The net photosynthetic rate (*Pn*), transpiration rate (*Tr*), and stomatal conductance (*Gs*) were measured using a LI-6400 portable photosynthesis system (Li-COR, Inc., Lincoln, NE, USA) from 10:00 to 11:00 a.m. The photosynthetically active radiation (PAR) at the leaf surface was 1400 ± 50 µmol m^−2^ s^−1^, the temperature of the leaf chamber was 25 ± 2 °C, and the ambient CO_2_ concentration was 400 ± 50 µmol mol^−1^.

Chlorophyll content: the 3rd–5th leaves at the tip of a plant were chosen for measurement using a SPAD 502 chlorophyll meter (Spectrum Technologies, Inc., Aurora, IL, USA).

### 4.5. Antioxidant Enzyme Activities and MDA Content

The plant antioxidative defense enzymes, i.e., superoxide dismutase (SOD, EC 1.15.1.1), catalase (CAT, EC 1.11.1.6)], lipid peroxidation, and total phenols were each determined by assay kit (Nanjing Jiancheng Bioengineering Institute, Nanjing, China) as described below.

Enzyme activity assay: To determine the activities of SOD and CAT, the samples (leaves and roots) were crushed in nitrogen gas and 4% (*w*/*v*) polyvinylpolypyrrolidone. Then, the powder was homogenized in a 2.0 mL solution of 100 mM potassium phosphate buffer (pH 6.8) containing 3 mM dithiothreitol, 0.1 mM ethylenediaminetetraacetic acid (EDTA), and 1.0 mM phenylmethylsulfonyl fluoride (PMSF). The suspension was centrifuged for 15 min at 15,000× *g* at 4 °C. The supernatant was collected to determine SOD activity and CAT activity using the corresponding detection kits (Nanjing Jiancheng Bioengineering Institute, Nanjing, China) following the manufacturer’s instructions. One unit of SOD activity was defined as the amount of enzyme required for 1 mg tissue proteins in 1 mL of a reaction mixture, with SOD inhibition rates up to 50% as monitored at 560 nm. The activities of SOD were demonstrated with U mg^−1^ proteins. One unit of CAT activity was defined as 1 mg tissue proteins consumed in 1 µmol H_2_O_2_ at 405 nm for 1 s. The activities of CAT were demonstrated with mg^−1^ proteins [60].

Oxidative damage to lipids was expressed as MDA content: the leaves and roots were crushed in nitrogen gas and lyophilized in 1.6 mL of 50 mM phosphate buffer (pH 7.8) containing 0.2 mM EDTA and 2% polyvinylpyrrolidone (PVP). The samples were vortexed three times at 5 min intervals before being centrifuged at 12,000× *g* for 20 min at 5 °C. The supernatant was gathered and MDA content was determined according to the instructions of the corresponding detection kits (Nanjing Jiancheng Bioengineering Institute, Nanjing, China) [60].

Total phenols: total phenols were determined using Folin–Ciocalteu reagent following the procedure described. Fresh plant samples in a centrifuge tube with a cover resulted in roughly 0.1 g of plant powder after being pounded with liquid nitrogen. Samples were homogenized with ethanol–H_2_O (5:3, *v*/*v*) and centrifuged at 4000× *g* for 10 min at 4 °C. Following this, a kit was used to test the absorption value at 760 nm.

### 4.6. Quantification of (–)-Loliolide

(–)-Loliolide was quantified by liquid extraction/solid-phase extraction, followed by a triple-quadrupole mass spectrometer (TQD, Waters Co., Milford, MA, USA) equipped with an electrospray ionization source operating in positive mode. Root samples of two biotypes for each weed species with four replicates were freeze-dried and ground with liquid nitrogen. The resulting powder (250 mg) was extracted with 10 mL of a MeCN (acetonitrile)-H_2_O-HOAC mixture (90:9:1, *v*/*v*/*v*), vortexed for 5 min at 25 °C. Then, NaCl was added and immediately vortexed for 1 min. The solution was centrifuged at 2800× *g* for 10 min and the supernatant was filtered with a 0.22 μm nylon syringe filter (Sterlitech, Kent, WA, USA) [34]. The filtrates were evaporated to dryness individually under vacuum and dry residues were dissolved in 50% aqueous methanol. The chromatographic separation conditions were as follows: an Acquity UPLC-BEH C18 column (50 mm × 2.1 mm, 1.7 μm) was used at 40 °C. The injection volume was 5 μL. The elution gradient, 0.0 min/90% A, 2–3 min/10% A and 4–5 min/90% A, was conducted with a binary solvent system consisting of 0.2% HOAC in H_2_O (solvent A) and MeCN (solvent B) at a constant flow rate of 0.3 mL min^−1^. Separation and stabilization were completed in 5.0 min. The typical conditions were as follows: capillary voltage 3.0 kV, source temperature 120 °C, and desolation temperature, 350 °C. (–)-Loliolide was quantified by regression analysis of the peak areas against standard concentrations [30].

### 4.7. Data Analysis

All data collected from experiments with three replicates were presented as means ± standard error (SE). Student’s *t*-test was conducted to compare significant differences on morphological, physiological, and biochemical parameters and relative fitness between resistant and susceptible biotypes for the six weeds tested.

Plant relative fitness contained two parts: viability fitness and fecundity fitness. The relative viability fitness was estimated as the plant total biomass, the relative fecundity fitness was calculated by seed biomass (HKW for each individual). Thus, the relative fitness (RF) was calculated as W_R_/W_S_, and the magnitude of the fitness cost (FC) was estimated as: FC = 1 − RF, where W is the quantitative estimation of a fitness trait by plant total biomass or seed biomass from resistant biotype (W_R_) and susceptible biotype (W_S_). For those weeds in which WR is higher than WS (W_R_/W_S_ > 1), this denotes a fitness advantage of the resistant over the susceptible biotype; if WR is lower than W_S_ (W_R_/W_S_ < 1) the viability and fecundity traits were integrated in the estimation of fitness cost. Fitness cost estimates range from 0.99 to 0, indicative of nearly lethal and negligible costs, respectively.

The variance contribution of each principal component was used as the weight, and the composite score model of the principal components was obtained by linear weighting of the principal component scores and the corresponding weights, which is the composite score model of ecological adaptability: Y = 0.292 Y1 + 0.240 Y2 + 0.178 Y3 + 0.0941 Y4 (Appendix A). The KMO test and Bartlett’s sphere test were used to determine whether factor analysis was applicable after the raw data were standardized (Z-score) to create new data. Since there was a strong correlation between the indicators and the main component analysis results, the KMO value was larger than 0.6 and the significance was lower than 0.05. Four main components could explain 80.404% of the total variance after the data underwent dimensionality-reduction analysis, showing that the principal component extraction was satisfactory (Appendix A). These four principal components reflected most of the information of the original variables. The 4 principal components were extracted instead of 22 indicators to comprehensively evaluate the adaptability of resistant and susceptible weeds.

All data analyses were performed with SPSS 26.0 for Windows (SPSS, Chicago, IL, USA). Figures were created using GraphPad prism 9.0.

## Figures and Tables

**Figure 1 plants-12-03158-f001:**
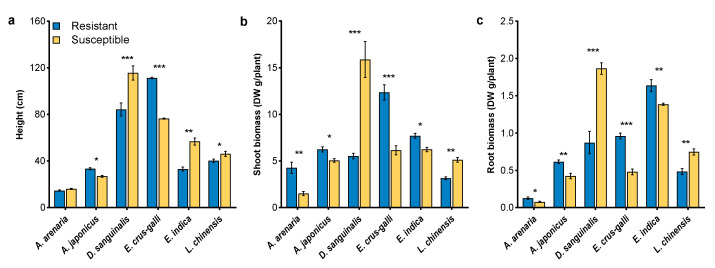
Morphology characteristics of herbicide-resistant and -susceptible weeds at the flowering stage. (**a**), plant height; (**b**), shoot biomass; (**c**), root biomass. Asterisks indicate significant difference between resistant and susceptible biotypes, Student’s *t*-test, * *p* < 0.05, ** *p* < 0.01, *** *p* < 0.001.

**Figure 2 plants-12-03158-f002:**
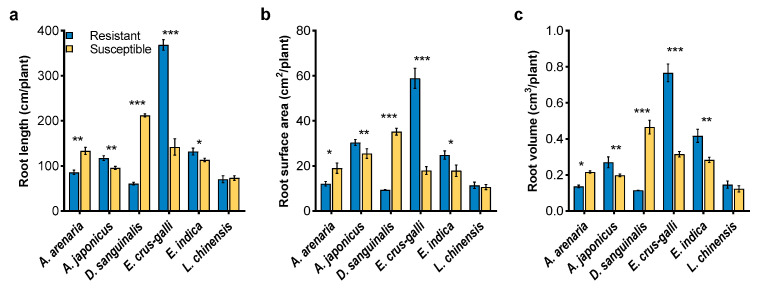
Root measurements of herbicide-resistant and -susceptible weeds at the seedling stage. (**a**), root length; (**b**), root surface area; (**c**), root volume. Asterisks indicate significant difference between resistant and susceptible biotypes, Student’s *t*-test, * *p* < 0.05, ** *p* < 0.01, *** *p* < 0.001.

**Figure 3 plants-12-03158-f003:**
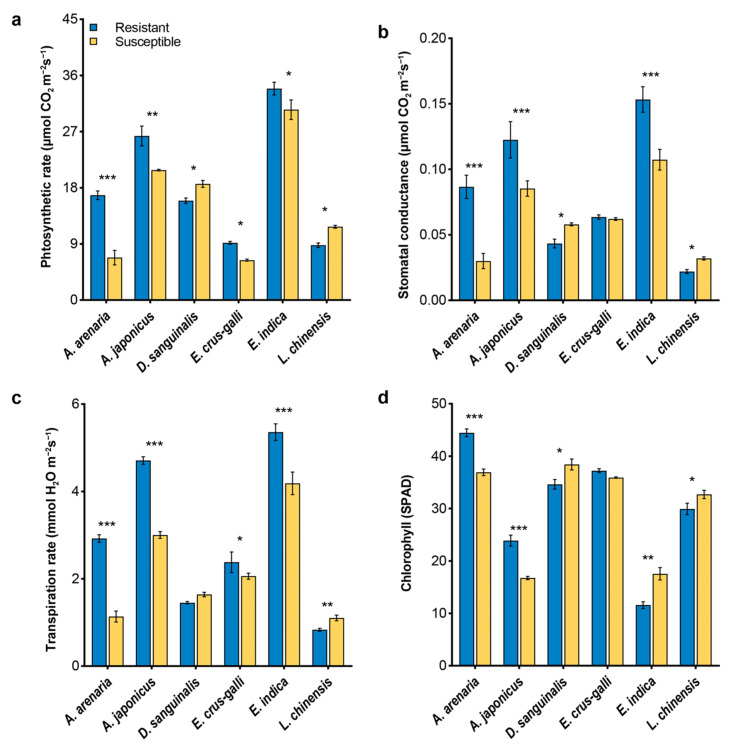
Photosynthetic parameters and chlorophyll concentrations of herbicide-resistant and sus−ceptible weeds during the flowering stage. (**a**), photosynthetic rate; (**b**), stomatal conductance; (**c**), transpiration rate; (**d**), chlorophyll. Asterisks indicate significant difference between resistant and susceptible biotypes, Student’s *t*-test, * *p* < 0.05, ** *p* < 0.01, *** *p* < 0.001.

**Figure 4 plants-12-03158-f004:**
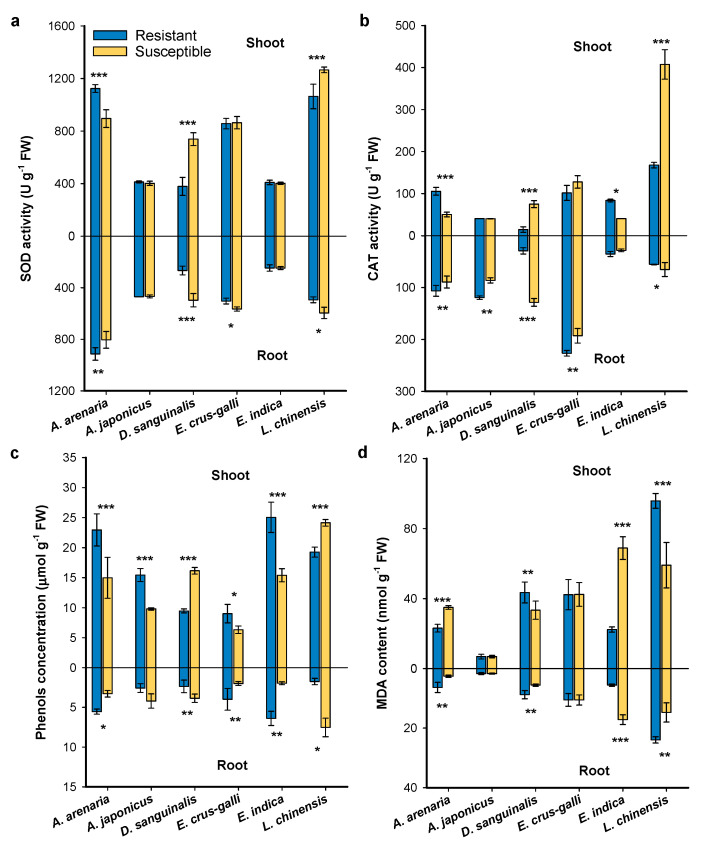
Effects of herbicide resistance on superoxide dismutase (SOD) activity (**a**), catalate (CAT) activity (**b**), total phenolic content (**c**), and malondialdehyde (MDA) content (**d**) of resistant and sus−ceptible weeds at the flowering stage. Asterisks indicate significant difference between resistant and susceptible biotypes. Student’s *t*-test, * *p* < 0.05, ** *p* < 0.01, *** *p* < 0.001.

**Figure 5 plants-12-03158-f005:**
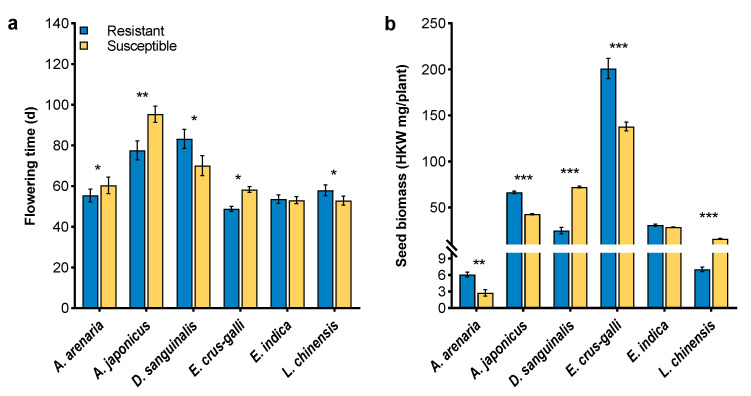
Flowering time (**a**) and seed biomass (**b**) in herbicide-resistant and -susceptible weeds. HKW means hundred kernel weight for each plant. Asterisks indicate a significant difference between resistant and susceptible biotypes, Student’s *t*-test, * *p* < 0.05, ** *p* < 0.01, *** *p* < 0.001.

**Figure 6 plants-12-03158-f006:**
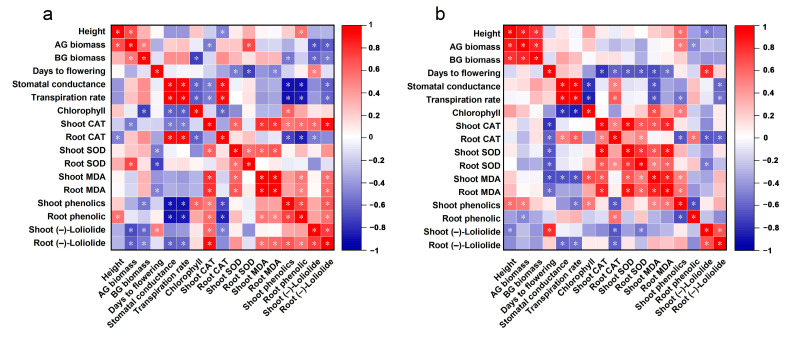
The correlations between the morphological, physiological, and biochemical indices of resistant and susceptible weeds. (**a**), resistant weeds; (**b**), susceptible weeds, Student’s *t*-test, * *p <* 0.05.

**Figure 7 plants-12-03158-f007:**
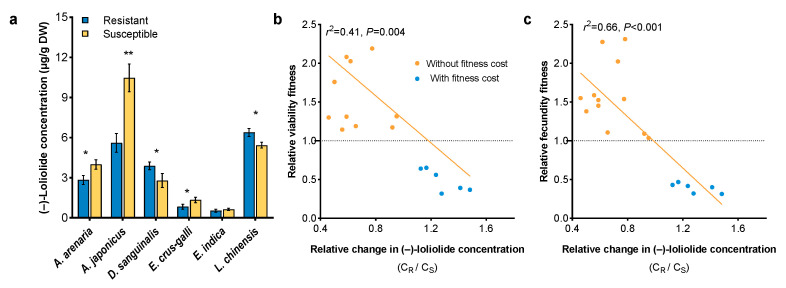
(–)-Loliolide concentration (**a**) in herbicide-resistant and -susceptible weeds, and relationships between the relative change in (–)-loliolide concentration (C_R_/C_S_) and relative viability fitness (**b**) and fecundity fitness (**c**). C_R_ means (–)-loliolide concentration for resistant biotype, C_S_ for susceptible biotype. Asterisks indicate significant difference between resistant and susceptible biotypes, Student’s *t*-test, * *p* < 0.05, ** *p <* 0.01.

**Figure 8 plants-12-03158-f008:**
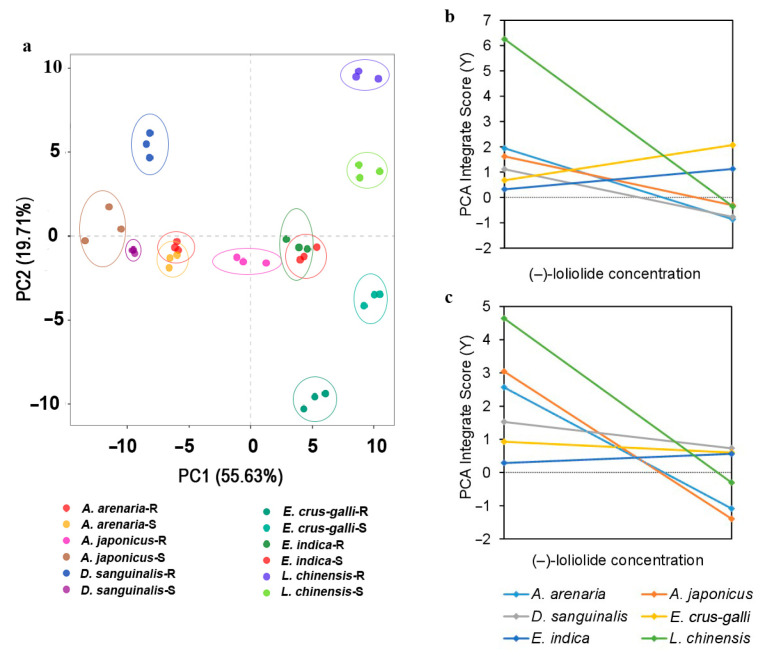
Plots of principal component analysis of weeds and variables (**a**), the relationship between (–)-loliolide concentration and ecological adaptability in herbicide−resistant (**b**), and -susceptible weeds (**c**).

## Data Availability

All data supporting the findings of this research are available within the paper and Appendix A.

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
