# Peer review of "Phytochemical Cue for the Fitness Costs of Herbicide-Resistant Weeds"

_plants, 2023, doi:10.3390/plants12173158_

Round 1

Reviewer 1 Report

"In this study, the researchers aimed to address the knowledge gap concerning the link between herbicide resistance costs and phytochemical cues in weed species. They conducted experiments on six herbicide-resistant weeds and their susceptible counterparts, evaluating viability, fecundity, and (–)-loliolide levels. Results showed significant differences in these parameters, with fitness costs observed in some resistant species but not in others. Correlations were found between morphological traits, (–)-loliolide levels, and MDA/phenol contents. Lower (–)-loliolide levels indicated stronger adaptability in E. crusgalli and E. indica. The study suggests the presence of fitness costs in some herbicide-resistant weeds, potentially regulated by (–)-loliolide's impact on vitality and fecundity."

I appreciate the time and effort that the authors have invested in conducting this research and drafting this manuscript. After a thorough review, I have identified some areas that could benefit from further improvement. I have provided detailed comments and suggestions in the hope that they will help the authors to strengthen their manuscript.

Comments:

Line 10 - Delete 'rapid evolution'

Line 11 - Delete 'multiple'

Line 12 - Delete 'experimentally'

Line 100 - Figure 1c. Double check the unit for root biomass.

Line 104 - Figure 2a,b,c. is this per plant?

Line 170-172 - Double check everything in this figure. The graph shows negative correlation. Use consistent identifying colors for resistant and susceptible types.

Line 263 - Delete 'adaptation'

Line 269-273 - Use lower case for herbicides names.

I hope that these general comments and suggestions are helpful.

The authors might contemplate removing instances of 'the data generated' from the main text, given its inherent relevance in a research manuscript.

Reviewer 2 Report

This was an interesting paper and appears to be important as well, offering some potential explanations about issues involved with herbicide resistance fitness differentials.  As I am a weed scientist and not a plant physiologist, I struggled with some aspects of the paper.  To make this paper understandable to as wide an audience as possible, I would suggest you use a few sentences in places just to explain some aspects a bit better.  I knew nothing about loliolide before reading this paper, so a bit more explanation of the background to what this chemical is thought to be involved with would be useful.  I would suggest citing the commentary paper by C.L. Frost (2023) as this helped me understand aspects of this work that were a mystery to me initially (C L Frost (2023). Information potential of an ubiquitous phytochemical cue. New Phytologist 238(5): 1749-1751.)  I am still unclear why it is necessary to put “(-)-“ in front of the word each time it is stated.  Presumably this is because it is an isomer of some sort, but surely even if it is, it would make the work much easier to read if this was shorted down to just loliolide, which I see many published papers have done.  It could be stated initially that only the isomer is thought to be important but it will be shortened down to loliolide for the rest of the paper.  It would be also helpful if a sentence is used to explain what SOD activity, CAT activity and MDA content signify for those readers who are not experts in plant physiology.  These abbreviations by the way were first mentioned on Lines 125 to 129, and had not been defined prior to this point.  They should be defined at the first mention within the paper, and preferably also explained somewhere in the paper, even if just briefly.  Likewise in the caption for Fig 4, which should stand alone from the text, they need defining there as well. 

In the abstract, I feel you need to make it clearer that herbicide resistance actually gave a number of these weed species a significant advantage with respect to the various parameters measured.  Currently on Lines 17 and 18 you merely say that there was no decrease in fitness for four of the six cases you studied, whereas in fact many of those showed the opposite to a fitness cost and there were advantages to their growth even though no herbicide had been applied, which is an interesting finding in itself.  There needed to be a bit more explanation of how these six particular cases were selected, presumably those for which seeds were readily available close to where the research was conducted?  It was unclear if all of these fitness costs and advantages had been previously published or whether some of this was novel information, so that needs to be clarified.  The Line 22 in the abstract was particularly misleading when it states “Therefore, fitness costs are evident for herbicide resistant plants but not universal.”  The data in this paper showed more examples of an advantage to the resistant weeds even in the absence of herbicides than examples of decreased fitness.  

There were issues with English throughout the paper.  Some of the main ones are as follows:

-          Throughout the paper, there are many times when the word chlorophyll is spelt incorrectly.

-          Throughout the paper, Echinochloa crus-galli or E. crus-galli is presented without a hyphen, which is not correct.

-          Line 32: “will often has” should be “will often have”

-          Line 40: “have frequently reported” should be “have frequently been reported”

-          Line 54: “in a less seed” should be “in less seed”

-          Line 55: “effect” should be “affect”

-          Line 65: “involve in phytochemical” should be “involve phytochemical”

-          Line 122: two different errors in spelling chlorophyll

-          Line 140: “later 2 and 13 days” should be “2 and 13 days later”

-          Line 142: “folds” should be “fold”

-          Lines 156-7: This sentence make no sense

-          Lines 158-9: This sentence makes no sense either.  Improved English could be: “However, the negative correlation between the photosynthetic rate of susceptible weeds and loliolide concentrations was significant (Figure 6).”

-          Line 74: The word “relationships” should be “correlations”

-          Line 199: “Evolutionary biology has believed that…” should be “Evolutionary biologists have often stated that herbicide resistance may lead to reduced fitness”

-          Lines 259- 260: This is another example of over-simplification of the findings;  rather than it not being universal for resistance to lead to fitness costs, you actually showed several examples of resistance leading to improved fitness even in the absence of herbicides.

-          Lines 271-273: Unless it is the start of a sentence, herbicide names should not start with a capital letter.

-          On Lines 272 and 274, the word “local” was used.  This tells readers nothing.  It would be better to use a term such as from eastern parts of China.  As all paddies were in Jiangsu Province of China, then you may as well say that, which is much better than “local”.

-          Line 275: “herbicides treatment” should be “herbicide treatments”

-          Line 278: “to ensure pollination” should be “to ensure no cross-pollination”

-          Line 279: “plastic buckets for the growth to maturity” should be “plastic pots and grown to maturity.”  The size of the pots should ideally be mentioned, ie volume of the pots.

-          Line 310: “six weeds” should be “six weed species”

-          Line 311: Should “Sterilized weeds” actually be “Surface-sterilized weed seeds”?  If the seeds had been sterilized, they would no longer be viable.

-          Line 314: “spaced” should be “sown”

-          Line 315: “deepth” should be “depth” or “deep”.  Also, weight of soil means many things depending on the moisture content, so it is more usual to state the volume of soil used.

-          Line 335: “among” should be “between”

-          Line 342: “3-5th leaves” should be “3rd-5th leaves”

-          Line 389: The final sentence on this line appears to make no sense, ie it appears to be incomplete at the end.
